# Effects of Gastrodin against Lead-Induced Brain Injury in Mice Associated with the Wnt/Nrf2 Pathway

**DOI:** 10.3390/nu12061805

**Published:** 2020-06-17

**Authors:** Chan-Min Liu, Zhi-Kai Tian, Yu-Jia Zhang, Qing-Lei Ming, Jie-Qiong Ma, Li-Ping Ji

**Affiliations:** 1School of Life Science, Jiangsu Normal University, No.101, Shanghai Road, Tongshan New Area, Xuzhou 221116, China; lcm9009@126.com (C.-M.L.); zzkisyai@163.com (Z.-K.T.); z774509@163.com (Y.-J.Z.); mingqinglei@jsnu.edu.cn (Q.-L.M.); 2College of Chemical Engineering, Sichuan University of Science and Engineering, Xuyuan Road, Zigong 643000, China; a1032042419@126.com; 3College of Physical Education, Jiangsu Normal University, No.101, Shanghai Road, Tongshan New Area, Xuzhou 221116, China

**Keywords:** gastrodin, lead, Wnt, Nrf2, inflammation, apoptosis

## Abstract

Gastrodin (GAS), the main phenolic glycoside extracted from *Gastrodia elata* Blume, exhibited potential neuroprotective properties. Here we examined the protective effects of GAS against lead(Pb)-induced nerve injury in mice, and explores its underlying mechanisms. Our research findings revealed that GAS improved behavioral deficits in Pb-exposed mice. GAS reduced the accumulation of p-tau and amyloid-beta (Aβ). GAS inhibited Pb-induced inflammation in the brain, as indicated by the decreased levels of pro-inflammatory cytokines, including tumor necrosis factor-a (TNF-α), cyclooxygenase-2 (COX-2). GAS increased the expression levels of NR2A and neurotrophin brain-derived neurotrophic factor (BDNF). GAS inhibited Pb-induced apoptosis of neurons in hippocampus tissue, as indicated by the decreased levels of pro-apoptotic proteins Bax and cleaved caspase-3. Furthermore, the neuroprotective effects of GAS were associated with inhibiting oxidative stress by modulating nuclear factor-erythroid 2-related factor 2 (Nrf2)-mediated antioxidant signaling. GAS supplement activated the Wnt/β-catenin signaling pathway and reduced the expression of Wnt inhibitor Dickkopf-1 (Dkk-1). Collectively, this study clarified that GAS exhibited neuroprotective property by anti-oxidant, anti-inflammatory and anti-apoptosis effects and its ability to regulate the Wnt/Nrf2 pathway.

## 1. Introduction

Gastrodin (GAS,4-hydroxybenzyl alcohol 4-O-β-D-glucopyranoside, PubChem CID: 115027) is the main phenolic glycoside extracted from the traditional Chinese herb “Tian ma” (Gastrodia elata Blume), which exhibited anti-oxidant, anti-inflammatory, anti-apoptosis, hepatoprotective and neuroprotective properties [1,2,3]. It is found that GAS can penetrate the blood-brain barrier and is quickly distributed throughout the brain after systemic oral administration [4]. Previous studies have indicated that GAS treatment alleviated memory deficits by increasing the expression of NR2A and GluR2 in the cerebellum of diabetic rats [5]. GAS treatment ameliorates subarachnoid hemorrhage induced neurological deficit, oxidative stress, inflammation and apoptosis by increasing Nrf2 activation and suppressing the expressions of interleukin (IL)-1β and tumor necrosis factor (TNF)-α [6].

The heavy metal lead is a ubiquitous environmental neurotoxicant that continues to be a leading environmental risk factor threatening public health [7,8]. Wnt signaling is critical for the normal function of the brain, which plays a key role in both nervous system development and adult synaptic plasticity [9]. Activation of the Wnt/β-catenin pathway can improve memory function and synaptic plasticity [10,11]. Previous studies have shown that Pb exposure causes cognitive deficits and synaptogenesis impairments by inhibiting the canonical Wnt pathway in vivo and in vitro [10,12]. To date, numerous studies have clarified that Pb can cause the impairments of spatial memory, synaptic transmission and synaptic plasticity via regulating the protein expressions of synaptophysin, NR2A receptor and neurotrophin brain-derived neurotrophic factor (BDNF) in brain of animals [12,13,14,15]. Pb exposure could cause oxidative stress, inflammation and apoptosis by inhibiting the activations of nuclear factor erythroid 2–related factor 2 (Nrf2) and homoxygenase-1 (HO-1) in brain of rats [1,16]. Prenatal Pb exposure could contribute to deficits in synaptic plasticity, which result in the amyloid-beta (Aβ) deposition, tau phosphorylation, mitochondrial dysfunction, caspase activation, and even cellular apoptosis [8,11,12,17]. However, to our knowledge, it has never been reported whether GAS could regulate the Wnt/Nrf2 pathway to protect the brain against Pb-induced injury.

Therefore, the objective of this study was to examine the protective effect of GAS against Pb-induced memory deficit, oxidative stress and inflammation in mice, and clarify the role of the Wnt/Nrf2 pathway in the action of GAS protection against Pb-induced brain injury.

## 2. Materials and Methods

### 2.1. Chemicals and Reagents

Gastrodin (98%) and lead acetate ((Pb(CH_3_COO)_2_)) were obtained from Sigma Chemical Co. (St. Louis, MO, USA). The p-tau, Aβ, Wnt7a, β-catenin, NR2A, BDNF, Nrf2, HO-1, NQO1, TNF-α, COX-2, Bcl-2, cytochrome C, cleaved caspase-3 and β-actin antibodies were supplied by Santa Cruz Biotechnology (CA, USA) and Abcam (Cambridge, MA, USA).

### 2.2. Animals and Ethics

Male ICR (mice (20 ± 1 g) were provided from Beijing HFK Bioscience CO., LTD (Beijing, China). All experiments process was approved by the respective university committees (No. IACUC-20.1.5) and performed according to national institutes of health guidelines for the care and use of animals and Chinese laws on the care of laboratory animals.

### 2.3. Experimental Design

The mice were kept for one week in a room with a circumambient temperature of 23 ± 1 °C, a 12 h dark/light cycle and relative humidity (55 ± 5)%. Then, the mice were randomly divided into four groups (15 mice/group); (1) Control group; (2) Pb group; (3) Pb+GAS(50 mg/kg b.w) group and (4) Pb+GAS(100 mg/kg b.w) group. In group (1), mice received equimolar acetate in drinking water in the form of Na acetate. In groups (2), (3) and (4), mice received an aqueous solution of lead acetate (Pb(CH_3_COO)_2_) at a concentration of 250 mg Pb/L of drinking water. In groups (3) and (4), mice were also supplied with GAS 50 or 100 mg/kg b.w. intragastrically once daily. The choice of Pb dose was based on previous reports [8,12]. The dose of GAS selected in this study was based on previously published data on the neuroprotective effect of GAS [1,6].

At the end of 4 weeks, mice were sacrificed by decapitation. Brains were collected immediately for future experiments.

### 2.4. Step-Down Test

The learning and memory ability was measured using the step-down test according to a previous report [18]. The step-down test training was performed 24 h after the final administration of GAS. The step-down test is a one-time stimulus avoidance response test. The latency and number of errors were recorded as memory test scores.

### 2.5. Golgi Stain

The brain was processed for Golgi staining according to the manufacturer’s instructions for the FD Rapid Golgi Stain kit (FD NeuroTechnologies, Inc., Columbia, MD, USA). For quantitative analysis, at least fifteen neurons from three animals of each group were analyzed. Spines were counted at high magnification (100 × oil objective). Spine density was calculated per 10 μm of dendritic length [10,15].

### 2.6. Biochemical Analysis

The malondialdehyde (MDA) concentration, total antioxidant capacity (TAC), the activities of superoxide dismutase (SOD) and glutathione (GSH) in the brain were analyzed by using commercial kits from Jiancheng Institute of Biotechnology (Nanjing, China) [8].

### 2.7. Western Blotting Analysis

The hippocampus-only protein expressions of the p-tau, Aβ, Wnt7a, β-catenin, NR2A, BDNF, Nrf2, HO-1, NQO1, TNF-α, COX-2, Bcl-2, cytochrome C, cleaved caspase-3 and β-actin were analyzed by Western blot according to the manufacturer’s guidelines (Bio/Rad, Hercules, CA, USA) [8,19]. The total protein samples for Western blotting were obtained by using a nuclear/cytoplasmic isolation kit (Beyotime Institute of Biotechnology, Beijing, China). Band intensities were quantified using Image J 1.42 software (NIH Bethesda, MD, USA). The vehicle control is set as 1.0. Data are expressed as mean ± S.E.M. and representative of five independent experiments (individual animals).

### 2.8. Statistical Analysis

Results were expressed as mean ± standard error (SE). The samples satisfied normality assumption; statistical significant differences between means were evaluated by Student’s t-test and one-way ANOVA followed by Tukey’s post hoc test for multiple comparisons. A value of *p* < 0.05 was considered statistically significant.

## 3. Results

### 3.1. GAS Alleviates Pb-Induced Memory Deficits and Reduction of Dendritic Spine Density of Mice

To assess the protective effects of GAS on Pb-induced memory deficits, the behavior of mice was measured using the Step-down test. As shown in Table 1, Pb exposure led to a marked reduction of latency in both the learning training (by 32.9%) and memory tests (by 36.5%), compared to those in the controls. Moreover, Pb exposure increased the number of errors in the learning training (by 76.8%) and memory tests (by 209.3%) compared to those in the controls. However, GAS treatment significantly improved the learning and memory ability of mice with a dose-dependent manner (*p* < 0.05). Moreover, the dendritic spine density was markedly decreased in the Pb group compared to the controls. GAS treatment effectively increased the dendritic spine density in the brain of mice (Figure 1 and Appendix A).

### 3.2. GAS Activated the Wnt Signaling Pathway in the Brain of Mice

Canonical Wnt/β-catenin-dependent signaling correlates with many neurological disorders, including synaptic dysfunction, memory deficit, neurodegeneration and Alzheimer’s disease [9,11]. We further measured the expressions of Wnt7a, β-catenin and the endogenous Wnt inhibitor Dickkopf-1 (Dkk-1) in the brain of mice. The results shown in Figure 2 demonstrate that Pb exposure reduced the expressions Wnt7a, β-catenin and increased Dkk-1 compared to the controls. However, these effects were blocked by GAS administration in the brain of mice (*p* < 0.05).

### 3.3. GAS Improved Hippocampal Plasticity and Neurotransmission of Mice

To evaluate the role of GAS treatment on the hippocampal plasticity and synaptic transmission of mice, the protein expressions of BDNF and NR2A were examined. The results showed that Pb decreased the protein expressions of BDNF and NR2A in the brain of mice compared to the controls (*p* < 0.05). However, the expression levels of these proteins in the brain of the Pb group were significantly up-regulated by GAS treatment (Figure 3).

### 3.4. GAS Inhibited Pb-Induced Oxidative Stress in the Brain of Mice

To evaluate the antioxidative effect of GAS, the MDA content and the activities of TAC and SOD were determined. As is showed in Table 2, compared with the control group, the content of MDA was elevated by 45.7% following Pb exposure, the activities of SOD and TAC decreased by 21.7% and 35.4%, respectively, which were partly reversed by GAS supplementation (*p* < 0.05).

### 3.5. GAS Regulated the Nrf2 Signaling Pathway in the Brain of Mice

We further examined the expression levels of Nrf2, HO-1 and NQO1 in the brain of mice. The results showed that Pb exposure decreased the nuclear translocation of Nrf2 and the protein expressions of HO-1 and NQO1 compared to the controls (*p* < 0.05). However, these effects were blocked by GAS administration in the brain of mice (Figure 4).

### 3.6. GAS Suppressed Pb-Induced Apoptosis in the Brain of Mice

To evaluate the anti-apoptosis function of GAS, we determined the expression of proteins related to apoptosis. As is shown in Figure 5, compared with the control group, the release of cytochrome c to the cytosol and the expression of cleaved caspase-3 were elevated, and the anti-apoptotic Bcl-2 decreased following Pb exposure, which was reversed by GAS treatment (*p* < 0.05).

### 3.7. GAS Suppressed Pb-Induced Inflammation in the Brain of Mice

To evaluate the anti-inflammatory function of GAS, we determined the level of the inflammatory cytokines in the brain of mice. As is shown in Figure 6, compared with the control group, the activity of NF-κB and the levels of inflammatory cytokines TNF-α and COX-2 were elevated after Pb exposure, which was reversed by GAS supplement (*p* < 0.05).

### 3.8. GAS Decreased Accumulation of P-Tau and Aβ in the Brain of Mice

The presence of excessive phosphorylated-tau (p-tau) and Aβ is able to induce inflammation and apoptotic neurodegeneration [8,12,18]. Here, the results indicated that Pb exposure resulted in the upregulation of p-tau and Aβ in the brain of mice as compared to the controls (*p* < 0.05). Interestingly, GAS treatment effectively decreased the deposition of these proteins in the brain of mice (Figure 7).

## 4. Discussion

This study focused on the neuroprotective role of GAS in a mouse model of Pb-induced injury. Meanwhile, the Wnt/β-catenin signaling pathway was activated by GAS treatment, which played a key role in mitigating the Pb-induced neurodegeneration, oxidative stress, inflammation and apoptosis. It is reported that Pb exposure has been linked to many hazardous effects on the brain of mice [8,14,18]. The data showed that Pb caused learning and memory impairments of mice (Table 1). It is noteworthy that GAS could improve memory deficits depression-like behaviors and neural stem cell proliferation in vivo and in vitro [20,21]. GAS is able to improve memory deficits and behavioral impairments in diabetic rats [5]. Our previous work showed that flavonoid fisetin ameliorated lead-induced neuroinflammation, apoptosis and memory impairment in mice [12]. In the present work, we revealed that GAS also improved the Pb-induced memory deficits (Table 1) and increased the dendritic spine density in mice (Figure 1). Thus, these results indicate that GAS exerts a neuroprotective property, alleviating Pb-induced cognitive impairment.

The Wnt/β-catenin signaling pathway is a vital factor in regulating hippocampal development and synaptogenesis [2,10,22]. The previous study indicated that Pb exposure could cause the synaptic impairments in the brain of mice via the Wnt7a/β-catenin signal pathway [10]. Lead exposure inhibited bone repair and cell differentiation by regulating Wnt/β-catenin signaling pathway [23]. The study demonstrated that GAS enhanced the neurogenesis and attenuated ischemis damage in a cerebral ischemic stroke model through the Wnt/β-catenin pathways [22]. GAS exhibited anti-inflammatory and anti-proliferation functions by activating the Wnt pathway [2]. Consistently, in this study, we observed that Pb exposure suppressed the protein expressions of Wnt7a and β-catenin and increased the protein expression of the Wnt/β-catenin signaling pathway inhibitor DKK-1 in the brain of mice. However, GAS supplement markedly decreased the DKK-1 expression and activated the Wnt/β-catenin signaling pathway (Figure 2). The data suggested that GAS treatment could activate the Wnt/β-catenin pathway to reduce the Pb-induced brain injury.

Wnt could regulate hippocampal plasticity and influences memory formation by triggering the transcription of Wnt target genes [22,24]. BDNF plays an important role in synaptic connectivity and ultrastructure [14]. A decrease of BDNF production has been found in many neurodegenerative diseases, including Alzheimer’s disease (AD) and Parkinson’s disease (PD) [25]. The activation of BDNF could be stimulated by the Wnt/β-catenin signaling pathway [26]. Recently, we revealed that Pb exposure caused impairments of synaptic plasticity in mice [12]. Pb exposure could decrease the level of BDNF which resulted in neuroinflammation, apoptotic neurodegeneration, neuronal plasticity, cognition and brain development [13,14]. NR2A, a subunit of NMDAR, is involved in synaptic plasticity, spine formation, synaptic transmission and adaptive neuronal responses [5,15]. NR2A downregulation may contribute to cognitive deficits, synaptic transmission impairments and development of dendritic spine change [15]. Several studies have demonstrated that Pb exposure decreases NR2A expression in the hippocampus of rats and in a cell culture system [15,27,28]. At present, our results also confirm that Pb exposure reduced the expressions of NR2A and BDNF, inevitably resulting in impairment of synaptic plasticity and behavior change [15,27]. The data showed that administration of GAS ameliorated motor learning impairments, neuroinflammation, degeneration and apoptosis by the up-regulation of BDNF and NR2A in diabetic rats [1,5]. The present study found that administration of GAS markedly increased the expression of these cognition-related proteins in hippocampus tissue of Pb group (Figure 3). This data demonstrated that GAS ameliorated Pb-induced impairments of synaptic plasticity by enhancing the expressions of cognition-related proteins.

Oxidative stress plays a fundamental role in the pathogenesis and development of Pb-induced neurotoxicity. Pb exposure induced the generation of reactive oxygen species (ROS), which further cause cellular structure damage and the peroxidation of cell membrane lipids [1,29]. Previous studies showed that Pb exposure could cause transcriptomic changes in human neural stem cells stimulating oxidative stress, which could associate with neurodevelopment in children [29]. Pb exposure caused spatial memory deficits and cell death by inducing oxidative stress in vivo and in vitro [30]. Nevertheless, GAS could alleviate motor performance in cerebral palsy patients and rescue cell death in macrophages by inhibiting oxidative stress [31]. GAS treatment increased blood-brain barrier permeability and attenuated subarachnoid hemorrhage induced brain damage by decreasing the oxidative stress and inflammation in rats [6]. At present, our results also found that Pb exposure increased MDA level and decreased the activities of SOD and TAC (Table 2), which could contribute to its pathogenesis by disrupting the pro-/antioxidant balance in cells and decreases the activities of several antioxidant enzymes [8,30]. However, GAS supplement decreased these markers of oxidative stress in the brain of Pb-exposed mice, which exhibited its neuroprotective properties against oxidative stress.

Nrf2, a redox-sensitive transcription factor, is closely related to many cell protection factors [1,30]. Under oxidative stress, Nrf2 translocates to the nucleus, binds to antioxidant response element (ARE), induces the expression of antioxidant genes, such as HO-1, NQO1 and GST [3,29,32]. Activated Wnt could inhibit neurogenic deficit and oxidative stress though the Nrf2 pathway in vivo and in vitro [33,34]. Pb exposure has been shown to decrease the expressions of Nrf2, HO-1 and other antioxidant molecules, which could weaken cell protection against oxidative stress [1,29,32]. GAS exerted anti-oxidant, anti-inflammatory, and anti-apoptotic effects in SH-SY5Y cells via the Nrf2 pathway [33]. Consistently, in this study, we observed that GAS treatment suppressed Pb-induced oxidative stress and restored the nuclear translocation of Nrf2 in the brain of mice. Moreover, the expression levels of HO-1 and NQO1 increased after GAS treatment (Figure 4). These results suggested that GAS treatment could activate the Nrf2 pathway to reduce the Pb-induced oxidative stress in the brain of mice.

Multiple studies have revealed that activation of the Wnt/β-catenin signaling pathway could inhibit neuronal apoptosis [35,36]. Recent studies indicated that Pb exposure caused spatial memory deficits and apoptosis in the brain via the Wnt/β-catenin pathway [37]. It has been shown that GAS supplement improved cognitive dysfunction and apoptosis in model of diabetic disease [1,5]. GAS also suppresses inflammation and apoptosis in H_2_O_2_-treated SH-SY5Y cells by regulating the Nrf2 pathway [3]. In the present study, the expression of anti-apoptotic proteins Bcl-2 decreased and the expressions of cytochrome c in cytosol and cleaved caspase-3 in the brain of the Pb group decreased following GAS administration (Figure 5), which might contribute to the improvement of memory and synaptic function.

Pb exposure could induce brain inflammation in vivo and in vitro, which is responsible for impairments of memory and synaptic transmission [1,12]. Wnt could promote dopaminergic neurorestoration in PD via the inflammatory Nrf2/HO-1 pathway [33,38]. Pb-induced ROS could activate NF-κB, which in turn promotes the release of inflammatory cytokines [12,30]. It has been reported that GAS exhibited neuroprotective property in animals by inhibiting inflammatory cytokine release in rat model of diabetic disease [1]. GAS treatment ameliorated cerebral ischemic injury by suppressing oxidative stress, inflammation and apoptosis by regulating the Nrf2 and NF-κB pathway [3]. GAS reversed depression-like behaviors and protected hippocampal neural stem cells by suppressing inflammatory response in a rat depression model [20]. GAS could reduce expressions of pro-inflammatory mediators and the release of pro-inflammatory cytokines in LPS-stimulated microglia [39]. In agreement with these results, we found that GAS treatment reduced the inflammatory cytokine release and NF-κB activity in the brain of the Pb group (Figure 6). These data demonstrated that the neuroprotective effects of GAS on Pb-induced brain dysfunction might also attribute to the reduction of inflammation.

Ample evidence showed that accumulation of Aβ and p-tau in the brain could cause memory impairments, inflammation and apoptosis [12,40]. The activation of the Wnt/β-catenin pathway could suppress the generation of Aβ and inhibit apoptosis in the mouse models of AD [40,41]. Previous studies have revealed that Pb exposure causes accumulation of Aβ and p-tau, further resulting in apoptosis and cognitive decline [8,12]. GAS alleviated memory dysfunction, inflammation and Aβ accumulation in the mouse models of AD [41]. GAS attenuated vascular dementia in a rat model by moderating the expressions of Aβ and p-tau in the brain [42,43]. It is worth noting that GAS treatment reduced the generation of Aβ and p-tau in the brain of mice compared with the Pb group (Figure 7). This data implies that GAS prevented Pb-induced neurotoxicity by decreasing the levels of p-tau and Aβ in hippocampus tissue of mice.

In summary, our research clarified that GAS administration markedly attenuated Pb-induced synaptic deficit, oxidative stress, inflammation and apoptosis via regulation of the Wnt/Nrf2 pathway. The neuroprotection of GAS warrants further investigation in our future research.

## Figures and Tables

**Figure 1 nutrients-12-01805-f001:**
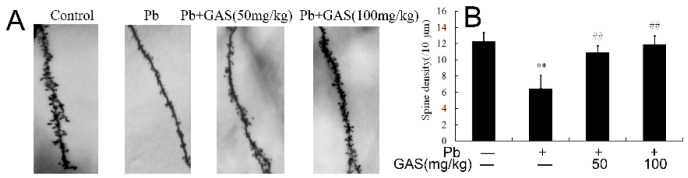
Gastrodin (GAS) increases dendritic spine density in the brain of mice. (**A**) Shifts of dendritic spine distribution in the brain; (**B**) density of dendritic spine in the brain. ^##^
*p*< 0.05, compared with the control group; ** *p* < 0.05, vs. Pb-treated group.

**Figure 2 nutrients-12-01805-f002:**
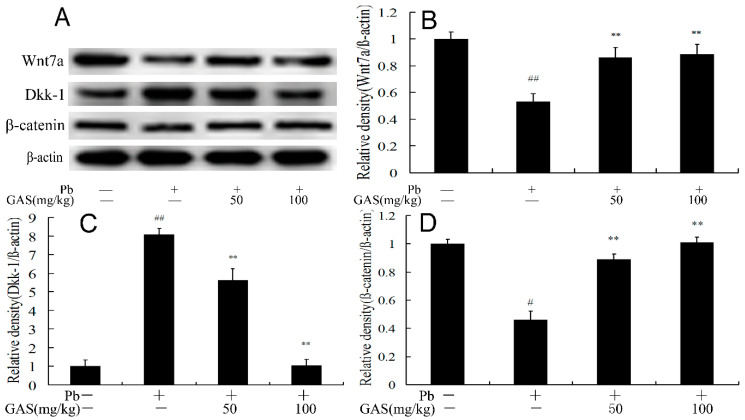
Gastrodin (GAS) activated the Wnt pathway in the brain of Pb-exposed mice. (**A**) Western blot analysis of the proteins of Wnt pathway in the brain; (**B**) relative density analysis of the Wnt7a protein bands; (**C**) relative density analysis of the Dkk-1 protein bands; (**D**) relative density analysis of the β-catenin protein bands. β-actin was probed as an internal control in relative density analysis. The vehicle control is set as 1.0. Data are expressed as mean ± S.E.M. and representative of five independent experiments (individual animals). ^##^
*p* < 0.05, compared with the control group; ** *p* < 0.05, vs. Pb-treated group.

**Figure 3 nutrients-12-01805-f003:**
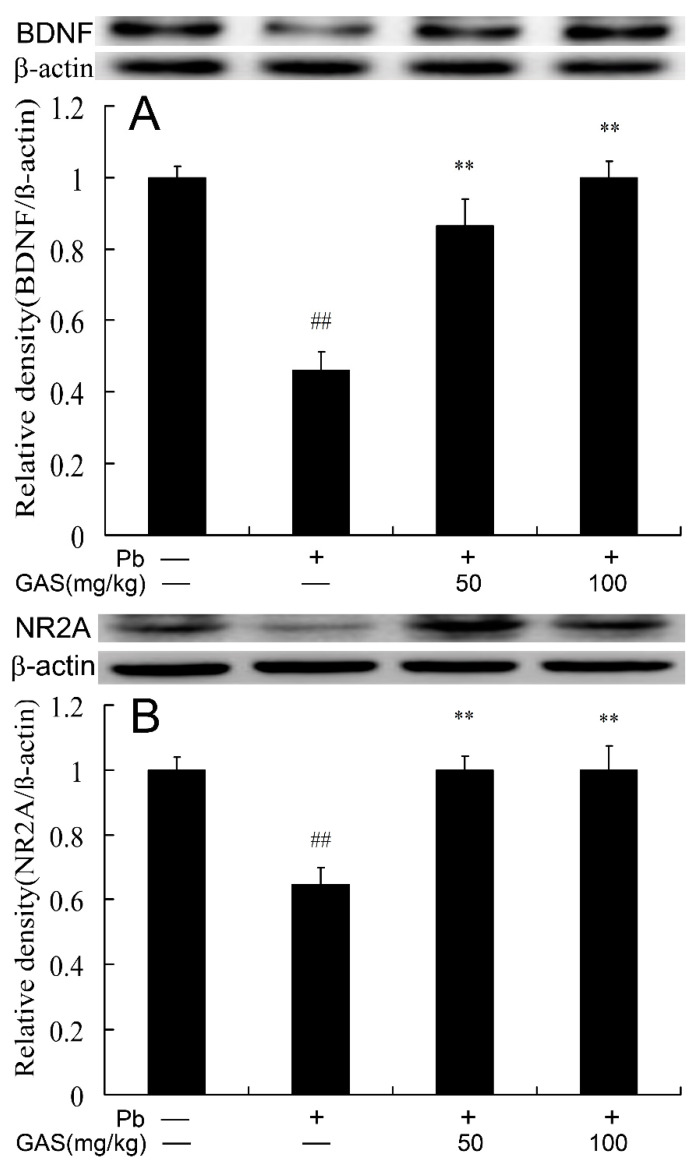
Gastrodin (GAS) alleviated Pb-induced synaptic dysfunction in the brain of mice. (**A**) Relative density analysis of the BDNF protein bands; (**B**) relative density analysis of the NR2A protein bands. β-actin was probed as an internal control in relative density analysis. The vehicle control is set as 1.0. Data are expressed as mean ± S.E.M. and representative of five independent experiments (individual animals). ^##^
*p* < 0.05, compared with the control group; ** *p* < 0.05, vs. Pb-treated group.

**Figure 4 nutrients-12-01805-f004:**
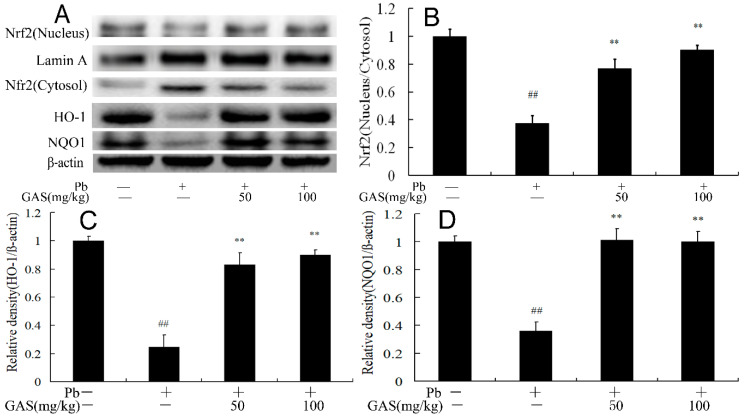
Gastrodin (GAS) increased activated Nrf2 pathway in the brain of Pb-exposed mice. (**A**) Western blot analysis of the proteins of Nrf2 pathway in the brain; (**B**) relative density analysis of the Nrf2 protein bands; (**C**) relative density analysis of the HO-1 protein bands; (**D**) relative density analysis of the NQO1 protein bands. β-actin was probed as an internal control in relative density analysis. The vehicle control is set as 1.0. Data are expressed as mean ± S.E.M. and representative of five independent experiments (individual animals). ^##^
*p* < 0.05, compared with the control group; ** *p* < 0.05, vs. Pb-treated group.

**Figure 5 nutrients-12-01805-f005:**
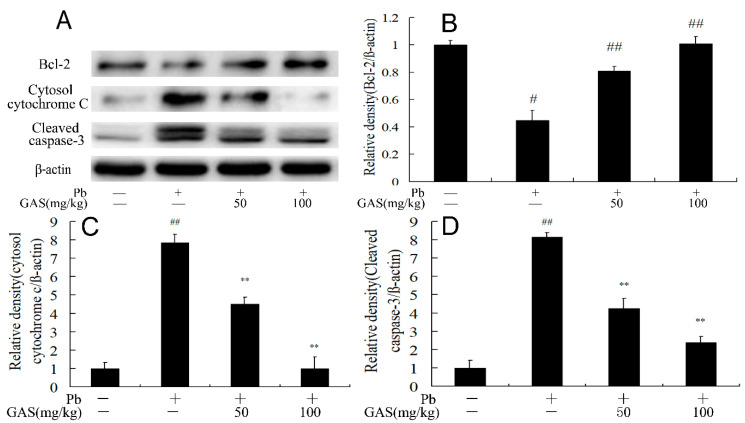
Gastrodin (GAS) inhibited Pb-induced apoptosis in the brain of mice. (**A**) Western blot analysis of the apoptosis-provoking proteins in the brain; (**B**) relative density analysis of the Bcl-2 protein bands; (**C**) relative density analysis of the cytochrome c in cytosol protein bands; (**D**) relative density analysis of the cleaved caspase-3 protein bands. β-actin was probed as an internal control in relative density analysis. The vehicle control is set as 1.0. Data are expressed as mean ± S.E.M. and representative of five independent experiments (individual animals). ^##^
*p* < 0.05, compared with the control group; ** *p* < 0.05, vs. Pb-treated group.

**Figure 6 nutrients-12-01805-f006:**
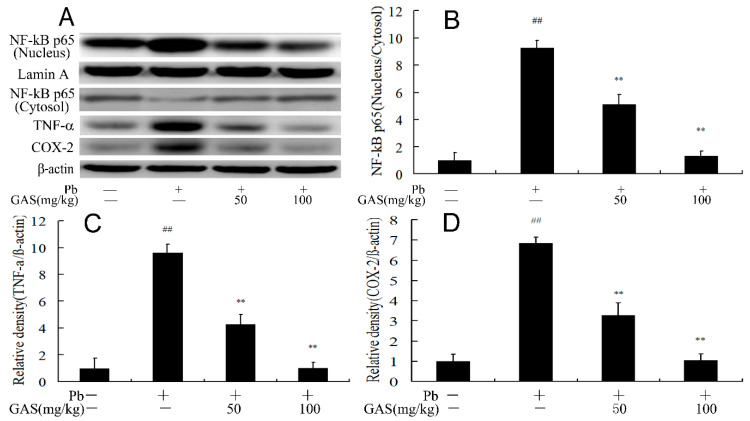
Gastrodin (GAS) inhibited Pb-induced inflammation in the brain of mice. (**A**) Western blot analysis of the apoptosis-related proteins in the brain; (**B**) relative density analysis of the NF-κB protein bands; (**C**) relative density analysis of the TNF-α protein bands; (**D**) relative density analysis of the COX-2 protein bands. β-actin was probed as an internal control in relative density analysis. The vehicle control is set as 1.0. Data are expressed as mean ± S.E.M. and representative of five independent experiments (individual animals). ^##^
*p* < 0.05, compared with the control group; ** *p* < 0.05, vs. Pb-treated group.

**Figure 7 nutrients-12-01805-f007:**
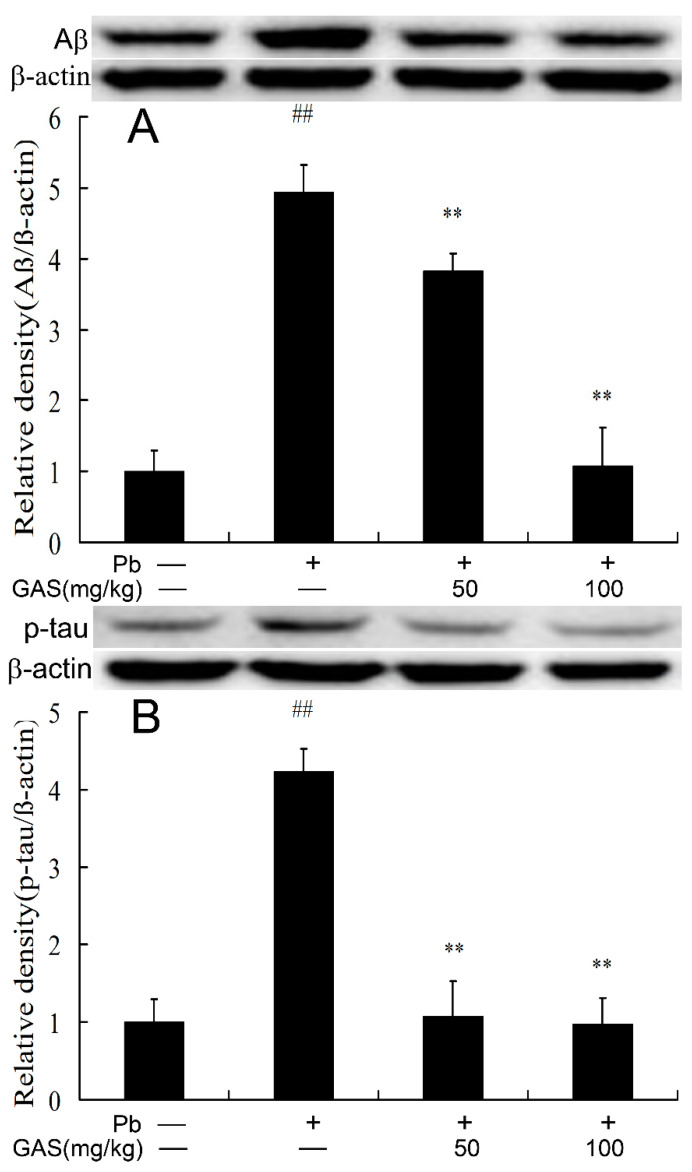
Gastrodin (GAS) reduced p-tau and Aβ accumulation in the brain of Pb-treated mice. (**A**) Relative density analysis of the Aβ protein in the brain; (**B**) relative density analysis of the p-tau protein bands. The vehicle control is set as 1.0. Total tau or β-actin were probed as an internal control in relative density analysis. The vehicle control is set as 1.0. Data are expressed as mean ± S.E.M. and representative of five independent experiments (individual animals). ^##^
*p* <0.05, compared with the control group; ** *p* < 0.05, vs. Pb-treated group.

**Table 1 nutrients-12-01805-t001:** Effects of gastrodin (GAS) on learning and memory abilities in lead-exposed mice in the step-down test.

	Latency (Seconds)		The Number of Errors	
Group	Learning training test	Memory test	Learning training test	Memory test
Control	81.08 ± 8.62	105.79 ± 10.93	2.46 ± 0.40	0.75 ± 0.10
Pb	54.41 ± 10.34 ^##^	67.14 ± 9.79 ^##^	4.35 ± 0.49 ^##^	2.32 ± 0.36 ^##^
Pb+GAS (50 mg/kg)	70.16 ± 7.34 **	83.23 ± 6.47 **	3.63 ± 0.38 **	1.21 ± 0.12 **
Pb+GAS (100 mg/kg)	79.18 ± 6.82 **	89.55 ± 7.16 **	2.96 ± 0.52 **	0.95 ± 0.14 **

Data are expressed as mean ± S.E.M. (*n* = 15). One-way ANOVA was used for comparisons of multiple group means followed by post hoc testing. ^##^
*p* < 0.05, compared with the control group; ** *p* < 0.05, vs. Pb-treated group.

**Table 2 nutrients-12-01805-t002:** Gastrodin (GAS) inhibited Pb-induced oxidative stress in the brain of mice.

	MDA (nM/mg prot)	SOD (U/mg prot)	TAC (U/mg prot)
Control	10.18 ± 1.03	324.15 ± 26.42	3.14 ± 0.16
Pb	14.83 ± 1.31 ^##^	253.69 ± 30.61 ^##^	2.03 ± 0.13 ^##^
Pb+GAS (50 mg/kg)	12.16 ± 1.08 **	298.95 ± 21.68 **	2.65 ± 0.22 **
Pb+GAS (100 mg/kg)	10.37 ± 1.12 **	319.06 ± 18.32 **	2.97 ± 0.17 **

Data are expressed as mean ± S.E.M. (*n* = 15). One-way ANOVA was used for comparisons of multiple group means followed by post hoc testing. ^##^
*p* < 0.05, compared with the control group; ** *p* < 0.05, vs. Pb-treated group.

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
