# Peer review of "Effects of Gastrodin against Lead-Induced Brain Injury in Mice Associated with the Wnt/Nrf2 Pathway"

_nutrients, 2020, doi:10.3390/nu12061805_

Round 1
Reviewer 1 Report
INTRODUCTION
1-The introduction highlights Pb, more than the proposed treatment by authors. Perhaps an inversion in the introduction text, starting with gastrodin and its effect may be more interesting.
lines 37 and 43, 45: brains => brain
line 44: cause
line 53: author must cite the via of administration.
line 61: complete the local of analysis.
METHODS
1-At various points in the manuscript, especially in the introduction and discussion, the authors cite the synaptic plasticity
involved with the Wnt / Nrf2 pathway. However, despite many data that indirectly indicate this involvement, the authors should present
photomicrographs of the brain/hipocampus immunostained with synaptophysin or synapsin, to, in fact, demonstrate, in situ, this correction
between synaptic plasticity or density in the brain with the Wnt/Nrf2 and their changes in the face of treatment with grastrodin.
Or any other morphological technique such as electron microscopy, light microscopy using silver impregnation, etc.
2- The author talks about using 40 animals (line 70) and later that he made 5 groups with 10 animals (line 77). Was it 40 or 50 mice? Or is the group number wrong?
3- Item 2.5: why didn't they measure glutathione (the brain's main antioxidant)?
4- Item 2.6: In the western blotting technique was used a brain pool of the animals in the group made? Or was used each brain individually? Were all 10 brains analyzed by each group? Explain.
line 79: mice
RESULTS
1- Table 1: Behavioral / memory tests with a small sample of animals (10) hardly result in parametric data. I suggest that the authors show the values ​​of each animal with the mean and standard error in the manuscript or as complementary material.
2- Figure 2: It is not clear the N used. How only the hippocampusonly was used in here?If ,in the other analyzes, the authors indicate that they used the brain (it is = diencephalon + telencephalon).
line 119: 3.3....in the brain
lines 120 and 121: the sentence is part of introduction
line 156 and 197: 3.6.....in the brain

Author Response
Response to Reviewer 1 Comments
Dear editor,
Thank you very much for your kind considerations on our MS (nutrients-794948) and arranging a timely review for our MS. And we would like to thank referees for critical comments and thoughtful suggestions. We have responded to these suggestions point by point, and revised the manuscript accordingly. All changes made to the text are indicated by red fonts so that they may be easily identified. Our responses to the reviewers' comments are as follows:
Point 1: INTRODUCTION
1-The introduction highlights Pb, more than the proposed treatment by authors. Perhaps an inversion in the introduction text, starting with gastrodin and its effect may be more interesting.
lines 37 and 43, 45: brains => brain
line 44: cause
line 53: author must cite the via of administration.
line 61: complete the local of analysis.
Response 1: Thank you for reviewing my article. According to the reviewer’s suggestion, we have revised the manuscript. We reversed the introduction text, starting with gastrodin and its effect.
lines 37 and 43, 45: brains => brain (Revised in Page 1-2)
line 44: cause(Revised in Page 2)
line 53: author must cite the via of administration. (Revised in Page 2)
line 61: complete the local of analysis. (Revised in Page 2)
Point 2: METHODS
1-At various points in the manuscript, especially in the introduction and discussion, the authors cite the synaptic plasticity involved with the Wnt / Nrf2 pathway. However, despite many data that indirectly indicate this involvement, the authors should present photomicrographs of the brain/hipocampus immunostained with synaptophysin or synapsin, to, in fact, demonstrate, in situ, this correction between synaptic plasticity or density in the brain with the Wnt/Nrf2 and their changes in the face of treatment with grastrodin. Or any other morphological technique such as electron microscopy, light microscopy using silver impregnation, etc.
Response 2: According to the reviewer’s suggestion, we have revised the manuscript. The dendritic spine density was analyzed by Golgi staining (Page 3, Fig.1).
Point 3: 2- The author talks about using 40 animals (line 70) and later that he made 5 groups with 10 animals (line 77). Was it 40 or 50 mice? Or is the group number wrong?
Response 3: According to the reviewer’s suggestion, we have revised the manuscript (Page 2).
Point 4: 3- Item 2.5: why didn't they measure glutathione (the brain's main antioxidant)?
Response 4: According to the reviewer’s suggestion, we have revised the manuscript. The determination method of glutathione is added to the method part (Page 3).
Point 5: 4- Item 2.6: In the western blotting technique was used a brain pool of the animals in the group made? Or was used each brain individually? Were all 10 brains analyzed by each group? Explain.
Response 5: According to the reviewer’s suggestion, we have revised the manuscript. Data are expressed as mean±S.E.M. and representative of five independent experiments (individual animals).
line 79: mice (Revised in Page 2)
Point 5: RESULTS, 1- Table 1: Behavioral/memory tests with a small sample of animals (10) hardly result in parametric data. I suggest that the authors show the values ​​of each animal with the mean and standard error in the manuscript or as complementary material.
Response 7: According to the reviewer’s suggestion, more data have been added. More mice have been used in behavioral tests (Tab.1). The values ​​of each animal with the mean and standard error have added as complementary material.
Point 5: 2- Figure 2: It is not clear the N used. How only the hippocampusonly was used in here?If ,in the other analyzes, the authors indicate that they used the brain (it is = diencephalon + telencephalon).
Response 7: According to the reviewer’s suggestion, we have revised the manuscript. The hippocampusonly protein was used in Western blotting analysis.
line 119: 3.3....in the brain(Revised in Page 4).
lines 120 and 121: the sentence is part of introduction(Revised in Page 4).
line 156 and 197: 3.6.....in the brain(Revised in Page 6)

Reviewer 2 Report
The manuscript (nutrients-794948) entitled "Effects of gastrodin against lead-induced brain injury in mice associated with the Wnt/Nrf2 pathway" describes the protective effects of gastrodin against lead induced nerve injury in mice, this work is interesting, but it needs a minor revision before it can be accepted for publication.
Change suggestions:
Line 53 - remove "and"
Line 76 - change "five" to "four"
Line 77 - remove the commas (standardize)
Line 92 - includes space between "capacity" and "TAC"
Line 96 - put "The" in lower case
Line 104 - change "Turkey" to "Tukey"?
Line 118 - remove the second full stop
Line 119 - check and correct the numbers "3.3" to "3.2"
Line 134 - check and correct the numbers "3.2" to "3.3"
Line 147 - check and correct the numbers "3.5" to "3.4"
Line 119 - check and correct the numbers "3.3" to "3.2"
Line 129 - change "DKK1"" to "DKK-1"
Line 155 - remove the second full stop
Line 156 - check and correct the numbers "3.6" to "3.5"
Line 170 - check and correct the numbers "3.3" to "3.6"
Line 184 - check and correct the numbers "3.4" to "3.7"
Line 197 - check and correct the numbers "3.6" to "3.8"
Line 229 - insert space between "pathway"and "(13)"
Line 232 - change "DKK1"" to "DKK-1"
Line 245 - insert space between "change" and "(9)"
Line 247 - change "exprosure" to "exposure"
Line 279 - put "The" in lower case
Line 289 - change "contrbute" to "contribute"
Line 290 - change "menmory" to "memory"
Line 303 - change "renoprotective" to neuroprotective"
Line 311 - change "tau" to "p-tau"
Line 345 - remove a period at the end of the reference
Line 353 - includes space between "Res. 2012, 1439"
Line 362 - "10.S Yousef, A.O.;” please delete "S" and includes space between "10" and "the name of the 1st author"
Line 395 - put 2014 in bold
Line 411 - include space between "Park," and "M"
Line 413 - includes space between "Cattaneo," and "E"
Line 417 - put full stop at end of reference
Line 420 - put full stop at end of reference
Line 423 - put full stop at end of reference
Line 431 - put 2016 in bold
Line 443 - put 2013 in bold
Line 452 - includes space between "Pathol. Res.Pract."
Line 457 - includes space between "Dis." and "2019"
Line 471 - change the references order (40 and 41)
Line 472 - put 2014 in bold
Line 476 - put 2018 in bold
Author Response
Response to Reviewer 2 Comments
Dear editor,
Thank you very much for your kind considerations on our MS (nutrients-794948) and arranging a timely review for our MS. And we would like to thank referees for critical comments and thoughtful suggestions. We have responded to these suggestions point by point, and revised the manuscript accordingly. All changes made to the text are indicated by red fonts so that they may be easily identified. Our responses to the reviewers' comments are as follows:
Point 1: The manuscript (nutrients-794948) entitled "Effects of gastrodin against lead-induced brain injury in mice associated with the Wnt/Nrf2 pathway" describes the protective effects of gastrodin against lead induced nerve injury in mice, this work is interesting, but it needs a minor revision before it can be accepted for publication. 

Response 1: Thank you for reviewing my article. According to the reviewer’s suggestion, we have revised the manuscript.
Point 2: Change suggestions……
Response 2:
According to the reviewer’s suggestion, we have revised the manuscript.
Line 53 - remove "and" (Revised in Page 2)
Line 76 - change "five" to "four" (Revised in Page 2)
Line 77 - remove the commas (standardize) (Revised in Page 2)
Line 92 - includes space between "capacity" and "TAC"(Revised in Page 3)
Line 96 - put "The" in lower case(Revised in Page 3)
Line 104 - change "Turkey" to "Tukey"?(Revised in Page 3)
Line 118 - remove the second full stop(Revised in Page 3)
Line 119 - check and correct the numbers "3.3" to "3.2"(Revised in Page 3)
Line 134 - check and correct the numbers "3.2" to "3.3"(Revised in Page 4)
Line 147 - check and correct the numbers "3.5" to "3.4"(Revised in Page 5)
Line 129 - change "DKK1"" to "DKK-1"(Revised in Page 3)
Line 155 - remove the second full stop (Revised in Page 6)
Line 156 - check and correct the numbers "3.6" to "3.5"(Revised in Page 6)
Line 170 - check and correct the numbers "3.3" to "3.6"(Revised in Page 7)
Line 184 - check and correct the numbers "3.4" to "3.7"(Revised in Page 7)
Line 197 - check and correct the numbers "3.6" to "3.8"(Revised in Page 8)
Line 229 - insert space between "pathway"and "(13)" (Revised in Page 9)
Line 232 - change "DKK1"" to "DKK-1"(Revised in Page 9)
Line 245 - insert space between "change" and "(9)" (Revised in Page 9)
Line 247 - change "exprosure" to "exposure"(Revised in Page 9)
Line 279 - put "The" in lower case(Revised in Page 10)
Line 289 - change "contrbute" to "contribute"(Revised in Page 10)
Line 290 - change "menmory" to "memory"(Revised in Page 10)
Line 303 - change "renoprotective" to neuroprotective"(Revised in Page 10)
Line 311 - change "tau" to "p-tau"(Revised in Page 11)
Line 345 - remove a period at the end of the reference (Revised in Page 11)
Line 353 - includes space between "Res. 2012, 1439"(Revised in Page 12)
Line 362 - "10.S Yousef, A.O.;” please delete "S" and includes space between "10" and "the name of the 1st author" (Revised in Page 12)
Line 395 - put 2014 in bold (Revised in Page 12)
Line 411 - include space between "Park," and "M"(Revised in Page 12)
Line 413 - includes space between "Cattaneo," and "E" (Revised in Page 12)
Line 417 - put full stop at end of reference(Revised in Page 12)
Line 420 - put full stop at end of reference(Revised in Page 12)
Line 423 - put full stop at end of reference(Revised in Page 12)
Line 431 - put 2016 in bold (Revised in Page 12)
Line 443 - put 2013 in bold (Revised in Page 12)
Line 452 - includes space between "Pathol. Res.Pract." (Revised in Page 12)
Line 457 - includes space between "Dis." and "2019" (Revised in Page 12)
Line 471 - change the references order (40 and 41) (changed the references order)
Line 472 - put 2014 in bold(Revised in Page 12)
Line 476 - put 2018 in bold(Revised in Page 12)

Reviewer 3 Report
This study investigated the protective effect of Gastrodin (GAS) on lead-induced brain injury. They found GAS treatment ameliorated lead-induced behavioral deficit in mice, the potential mechanisms include rescue of Wnt and Nrf2 signaling, inhibition of inflammation and oxidative stress, and inhibition of apoptosis. This study provides potential novel method for treatment of lead toxicity for the brain.
There are some concerns that need to be addressed:
- Result 3.1, table 1, looks like 100 mg/kg has a better outcome than 50 mg/kg, is there any significant difference in “number of errors” between the two doses of GAS? In other words, is there any doses response of GAS?
- Figure 3B Nrf2 quantification doesn’t match the Figure 3A images very well.
- Figure 4, Western blot for cytosol cytochrome C, how was the cytochrome C isolated from mice brain? This should be detailed in “Methods”.
- Figure 5, the authors measured NF-kB in nucleus, again, the Method of nuclei protein isolation from mice brain should be provided.
- Are there published reports about A-beta and p-Tau accumulation caused by lead-exposure? The accumulation is mainly in hippocampus as lead can induce memory deficit?
- In a study by Wu Yong etc (Planta Med, 2009 Aug;75(10):1112-7.), Gastrodin has been shown to protect against lead-induced synaptic deficit in rat. The authors should cite this paper as a supporting evidence of GAS’s neuroprotection effect.
Author Response
Response to Reviewer 3 Comments
Dear editor,
Thank you very much for your kind considerations on our MS (nutrients-794948) and arranging a timely review for our MS. And we would like to thank referees for critical comments and thoughtful suggestions. We have responded to these suggestions point by point, and revised the manuscript accordingly. All changes made to the text are indicated by red fonts so that they may be easily identified. Our responses to the reviewers' comments are as follows:
Point 1: This study investigated the protective effect of Gastrodin (GAS) on lead-induced brain injury. They found GAS treatment ameliorated lead-induced behavioral deficit in mice, the potential mechanisms include rescue of Wnt and Nrf2 signaling, inhibition of inflammation and oxidative stress, and inhibition of apoptosis. This study provides potential novel method for treatment of lead toxicity for the brain. 

Response 1: Thank you for reviewing my article. According to the reviewer’s suggestion, we have revised the manuscript.
Point 2: Result 3.1, table 1, looks like 100 mg/kg has a better outcome than 50 mg/kg, is there any significant difference in “number of errors” between the two doses of GAS? In other words, is there any doses response of GAS?
Response 2: According to the reviewer’s suggestion, we have revised the manuscript. We are analyzed the data and found that there is the doses response of GAS (Page 3).
Point 3: Figure 3B Nrf2 quantification doesn’t match the Figure 3A images very well.
Response 3: According to the reviewer’s suggestion, we have revised the Figure 3.
Point 4: Figure 4, Western blot for cytosol cytochrome C, how was the cytochrome C isolated from mice brain? This should be detailed in “Methods”.
Response 4: According to the reviewer’s suggestion, we have revised the manuscript. The cytochrome C isolated from mice brain used a nuclear/cytoplasmic isolation kit (Beyotime Institute of Biotechnology, Beijing, China). We supplement the detailed operation steps in the material (Page 3).
Point 5: Figure 5, the authors measured NF-kB in nucleus, again, the Method of nuclei protein isolation from mice brain should be provided.
Response 5: According to the reviewer’s suggestion, we have revised the manuscript. The cytochrome C isolated from mice brain used a nuclear/cytoplasmic isolation kit (Beyotime Institute of Biotechnology, Beijing, China). We supplement the detailed operation steps in the material (Page 3).
Point 6: Are there published reports about A-beta and p-Tau accumulation caused by lead-exposure? The accumulation is mainly in hippocampus as lead can induce memory deficit?
Response 6: According to the reviewer’s suggestion, we have revised the manuscript. Our previous studies and others have shown that lead exposure can cause A-beta and p-Tau accumulation in hippocampus. The lead accumulation of A-beta and p-Tau in hippocampus can induce memory deficit [1-4].
- Liu, C.M.; Yang, W.; Ma, J.Q.; Yang, H.X.; Feng, Z.J.; Sun, J.M.; Cheng, C.; Jiang, H. Dihydromyricetin inhibits lead-induced cognitive impairments and inflammation by the adenosine 5’-monophosphate-activated protein kinase pathway in mice. Agric. Food Chem. 2018. 66, 7975-7982.
- Yang, W.; Tian, Z.K.; Yang, H.X.; Feng, Z.J.; Sun, J.M.; Jiang, H.; Cheng, C.; Ming, Q.L.; Liu, C.M. Fisetin improves lead-induced neuroinflammation, apoptosis and synaptic dysfunction in mice associated with the AMPK/SIRT1 and autophagy pathway. Food Chem. Toxicol. 2019, 134, 110824.
- Bihaqi, S.W.; Alansi, B.; Masoud, A.M.; Mushtaq, F.; Subaiea, G.M.; Zawia, N.H. Influence of early life lead (Pb) exposure on α-synuclein, GSK-3β and caspase-3 mediated tauopathy: Implications on Alzheimer's disease. Alzheimer Res. 2018, 15, 1114-1122.
- Chin-Chan, M.; Cobos-Puc, L.; Alvarado-Cruz I; Baya,r M.; Ermolaeva, Early-life Pb exposure as a potential risk factor for Alzheimer's Disease: Are there hazards for the mexican population? J. Biol. Inorg Chem. 2019, 24, 1285-1303.
Point 7: In a study by Wu Yong etc (Planta Med, 2009 Aug;75(10):1112-7.), Gastrodin has been shown to protect against lead-induced synaptic deficit in rat. The authors should cite this paper as a supporting evidence of GAS’s neuroprotection effect.
Response 7: According to the reviewer’s suggestion, we have revised the manuscript. This paper (Wu Yong et al., Planta Med, 2009 Aug; 75(10):1112-7.) has been cited as a supporting evidence of GAS’s neuroprotection effect (Reference 21).

Round 2
Reviewer 1 Report
The authors followed the suggestions made by me. The text is more objective and clear. In that sense, I recommend publishing the manuscript in the Nutrients journal.